# SDC-Net++: End-to-End Crash Detection and Action Control for Self-Driving Car Deep-IoT-Based System

**DOI:** 10.3390/s24123805

**Published:** 2024-06-12

**Authors:** Mohammed Abdou Tolba, Hanan Ahmed Kamal

**Affiliations:** Department of Electronics and Communications Engineering, Faculty of Engineering, Cairo University, Cairo 12613, Egypt

**Keywords:** autonomous driving, deep learning, computer vision, multitask learning, crash detection, path planning, automatic emergency braking, camera-cocoon, IoT, system

## Abstract

Few prior works study self-driving cars by deep learning with IoT collaboration. SDC-Net, which is an end-to-end multitask self-driving car camera cocoon IoT-based system, is one of the research areas that tackles this direction. However, by design, SDC-Net is not able to identify the accident locations; it only classifies whether a scene is a crash scene or not. In this work, we introduce an enhanced design for the SDC-Net system by (1) replacing the classification network with a detection one, (2) adapting our benchmark dataset labels built on the CARLA simulator to include the vehicles’ bounding boxes while keeping the same training, validation, and testing samples, and (3) modifying the shared information via IoT to include the accident location. We keep the same path planning and automatic emergency braking network, the digital automation platform, and the input representations to formulate the comparative study. The SDC-Net++ system is proposed to (1) output the relevant control actions, especially in case of accidents: accelerate, decelerate, maneuver, and brake, and (2) share the most critical information to the connected vehicles via IoT, especially the accident locations. A comparative study is also conducted between SDC-Net and SDC-Net++ with the same input representations: front camera only, panorama and bird’s eye views, and with single-task networks, crash avoidance only, and multitask networks. The multitask network with a BEV input representation outperforms the nearest representation in precision, recall, f1-score, and accuracy by more than 15.134%, 12.046%, 13.593%, and 5%, respectively. The SDC-Net++ multitask network with BEV outperforms SDC-Net multitask with BEV in precision, recall, f1-score, accuracy, and average MSE by more than 2.201%, 2.8%, 2.505%, 2%, and 18.677%, respectively.

## 1. Introduction

Autonomous driving is a dream come true. Despite the restricted laws preventing fully automated vehicles, researchers and scientists do not spare any efforts in the field to prove its stability as a counter-push against these laws. The US department adopted six levels of driving automation [1] as shown in Figure 1 ranging from 0 (fully manual) to 5 (fully autonomous). These levels can be categorized into two categories: in the first, a human driver **must** monitor the surrounding environment, Levels (0–2), and in the second, an automated system monitors the environment, Levels (3–5).

**Level 0—No Driving Automation** is the level at which the driver fully controls the vehicle. **Level 1—Driver Assistance** is the level at which the adaptive cruise control (ACC) [2] system is already integrated into the vehicle, keeping the ego-vehicle at a safe distance from the up-front vehicle; however, the human driver monitors other control actions like steering and braking. **Level 2—Partial Automation** is the level at which the advanced driver assistance system (ADAS) [3] is integrated, so the vehicle controls both steering and acceleration/deceleration. Still, the driver can take control of the vehicle at any time. Tesla [4] and General Motors (GM) [5] are the largest companies in the field that achieve supercruise systems as Level 2. **Level 3—Conditional Driving Automation** is the level at which environmental detection capabilities are integrated, and the vehicle can take control actions but with human driver override, who takes control of actions in case the system executes wrongly. **Level 4—High Automation** is the level at which a failure system is integrated so that the vehicle can perform control actions and, in case of failure, a system can perform the right actions without human–driver interaction; however, the driver override is still an option. The vehicle can be operated in self-driving mode within low-speed and limited areas like urban areas in which the vehicle speed does not exceed 30 kmph; this is also geofencing. Volvo [6], Waymo [7], and Baidu [8] are the most significant companies at this level. **Level 5—Full Driving Automation** is the level at which human attention and interaction are not needed at all; the vehicle will take control actions in both urban and highway areas. Our proposed system targets this level.

The self-driving car pipeline consists of three main phases, as shown in Figure 2: Perception, Planning, and Control. **Perception** of the environment is the main building block of autonomous driving. Perception is implemented by collecting sensor readings from different sensors like lidars, radars, camera sensors, and more, and then fusing these sensor readings to achieve robust information about the surroundings [9]. Perception is decomposed into two sub-blocks: detection and localization. The detection part aims to classify and detect all the concerned objects around the vehicle by applying some sophisticated algorithms: lane detection, traffic light detection and classification, and object detection and tracking. The localization part aims to localize the ego vehicle in the world coordinates against the other objects with the help of a high-definition HD-map. **Path planning** [10] is one of the main blocks of the autonomous driving pipeline that is integrated into the system to identify all relevant control actions. Path planning is considered a multistage step that can be decomposed into the following sub-blocks. **Route Planning**: Also called global planning, route planning aims to identify the possible routes to reach the desired position by the HD-map assistance; **Trajectory Prediction** [11,12] benefits from the perception of the surrounding objects to identify all the possible trajectories. Trajectory prediction is made based on applying different motion models for the classified and detected objects in which there is a motion model for the cars, a different one for the pedestrians, etc. **Trajectory Planning** [13] computes and plans the trajectory based on the perception of the environment. After that, the ego vehicle plans its trajectory safely. **Control** [14,15] is responsible for taking the relevant and safe actions based on the trajectory planning step. It controls the ego-vehicle’s manipulated variables: throttle, steer, and brake. The control step takes a list of waypoints (trajectories) and target velocities generated by the planning step passing through controller algorithms like PID or MPC, which calculate how much to steer, accelerate, or brake to meet the target trajectory.

In this paper, we introduce the SDC-Net++ system as an extension to our previous research (SDC-Net) system. SDC-Net++ is an end-to-end system that aims to make vehicles navigate autonomously through the predicted action control: throttle, maneuver, and brake, in addition to a crash detection feature not only to detect crashes but also to help in navigation. Similar to SDC-Net, our dataset is created using the CARLA simulator, serving the main goal of having path planning, automatic emergency braking, and crash detection functionalities which are the main building blocks to set up a self-driving car. SDC-Net++ takes the input of four cameras installed around the ego-vehicle (camera cocoon) covering 360° around the vehicle. Then, as with SDC-Net, SDC-Net++ applies different input representations: front camera, panorama, and bird’s eye views. Both SDC-Net and SDC-Net++ apply a multitask network: in SDC-Net, this comprises crash classification, whose aim is to classify whether there is a crash or not in every scene, and an action control network, whose aim is to take precise control actions: throttle, brake, and maneuver. However, SDC-Net++ replaces the classification network with a detection network in which bounding boxes prediction is the output instead of binary classification. Both SDC-Net and SDC-Net++ are attached to IoT platforms that can communicate with other vehicles and share the crash information gathered.

The remainder of the paper is composed as follows: Section 2 includes a literature review on self-driving cars including our previous paper (SDC-Net). Section 3 describes the main contributions showing the SDC-Net++ system and the network architecture. Dataset adaptations and analysis are described in Section 4. Results and the conducted experiments are shown in Section 5 comparing crash detection only and multitask results. Section 6 elaborates meaningful comments on the conducted experiments and then discusses comparing SDC-Net++ with our previous network SDC-Net. A summarized conclusion is shown in Section 7.

## 2. Literature Review

No vehicle can drive itself, but the following come the closest. There is no fully autonomous vehicle for sale today, but some automakers are advancing the field. Over the past few years, a few cars have gone on sale with driver assistance features that take much of the burden off the driver. These systems alleviate driver fatigue by assisting with steering and acceleration. This technology is beneficial for drivers with long commutes, and it can come in handy during road trips requiring lots of driving. Much research has been conducted to achieve self-driving vehicles as soon as possible. Sallab et al. proposed a framework in [16] to achieve autonomous driving and then proved the success of this framework in [17] when they applied their proposed algorithm on the TORCS simulator [18]. Autonomous driving using imitation learning is also tackled in [14], which is inspired by Intel [19].

Loc et al. [20] introduced a fast and accurate 3D multi-object detection framework using bird’s eye view (BEV) representation. This method outperformed other techniques on the KITTI dataset, achieving a frame rate of 62 FPS. Hu et al. [21] utilized pseudo-LiDAR for monocular 3D object detection, integrating image segmentation with geometric constraints to enhance depth prediction. Zou et al. [22] compared Camera model-Based Feature Transformation (CBFT) and Camera model-Free Feature Transformation (CFFT) for BEV Semantic Segmentation. They introduced a Hybrid Feature Transformation module (HFT) to combine the strengths of both approaches. Gong et al. [23] proposed GitNet, a two-stage Geometry PrIor-based Transformation framework for BEV semantic segmentation, incorporating geometry-guided pre-alignment and a ray-based transformer.

BEVFormer [24] uses spatiotemporal transformers to learn unified BEV representations for various autonomous driving perception tasks. BEVSegFormer [25], another transformer-based approach, focuses on BEV semantic segmentation using multiple camera setups. Natan and Miura [26] introduced an end-to-end multi-task learning approach for BEV, enabling simultaneous perception and control tasks. Xie et al. [27] introduced $M^{2}BEV$, efficiently converting multi-view 2D image features into 3D BEV features. CoBEVT [28] is a transformer-based framework for multi-agent, multi-camera perception, achieving state-of-the-art performance in BEV semantic segmentation on the OPV2V and OpenCDA datasets.

V2VNet [29] enables vehicle-to-vehicle (V2V) communication, allowing vehicles to exchange information within a 70 m radius. COOPERNAUT [30] is an end-to-end model for vision-based cooperative driving, using point-based representations for cross-vehicle perception. CoBEVT [28], which is a multi-agent multi-camera perception framework, generates BEV map predictions cooperatively. This framework is a transformer-based architecture with a fused axial attention module (FAX). Spatial interactions are captured locally and globally across views and agents, achieving state-of-the-art performance when tested on OPV2V [31] for BEV semantic segmentation. OPV2V is a large-scale V2V perception dataset that depends on the cooperation between OpenCDA [32] and CARLA [33], where OpenCDA is a simulator that aims to develop and test Cooperative Driving Automation (CDA) systems. V2VNet [29] is a V2V communication system that covers a 70 m radius of vehicle connectivity, so vehicles can receive and publish information within this area. A spatial graph neural network (GNN) is used to collect information from all of the connected self-driving cars. It is also tested on V2X-Sim. COOPERNAUT [30] is an end-to-end model that cooperatively drives self-driving cars based on a cross-vehicle perception. It converts lidar information into a point-based representation that can be transmitted wirelessly between the connected vehicles. The authors also developed a CARLA-based framework AUTOCASTSIM with challenging accident-prone scenarios.

Wang et al. proposed DeepAccident [34], which is the first large-scale V2X autonomous driving dataset featuring various collision accident scenarios. End-to-end motion and accident prediction tasks are also introduced with metrics for evaluating prediction accuracy. DeepAccident includes sensor data and annotation labels from four vehicles and the surrounding infrastructure for each scenario, enabling V2X research in perception and prediction. Additionally, DeepAccident becomes a direct safety benchmark for autonomous driving algorithms and a supplementary dataset for single-vehicle and V2X perception research in safety-critical scenarios.

SDC-Net [35], which is an end-to-end multitask deep neural network, was designed to navigate safely by performing crash avoidance, path planning, and automatic emergency braking. Abdou et al. proposed a full system that combines deep learning with IoT to achieve self-driving cars; they built their dataset using the CARLA simulator due to the lack of real benchmark datasets. There was an initial trial in merging IoT with deep learning in [36]; however, it tackled only crash avoidance functionality. SDC-Net provided a comparative study between different input representations: front camera only, panorama, and BEV to be applied on a single crash avoidance-only task, a single path planning and AEB-only task, or multitask. As expected, they concluded that the multitask BEV experiment outperforms other ones in precision, recall, f1-score, accuracy, throttle MSE, steer MSE, and brake MSE.

## 3. System Methodology

This section tackles system methodology identifying the main modifications of the SDC-Net system, especially in the detection network. Figure 3 shows the SDC-Net++ system architecture in which the main modification is in the crash detection network, which is green. However, the other blocks are kept to formulate a comparative study between SDC-Net and our proposed system SDC-Net++.

### 3.1. Multitask Deep Neural Network

The multitask deep neural network, inspired by [35] with some modifications, starts with an input normalization phase to ensure a zero mean and one standard deviation. The **feature extraction part** for the multitask network is made up of 10 convolution layers, the first having kernel size (5,5), while the remaining 7 layers have kernel sizes (3,3). The first 4 convolution layers have 32 and 64 filters for every two consecutive layers. The next 3 convolution layers have 128, 64, and 128 filters, while the last 3 convolution layers have 256, 128, and 256 filters. Every convolution layer is followed by a ReLU activation function introducing nonlinearities. The first and second convolution layers are followed by a max-pooling layer with a (2,2) kernel size, while the third and fourth convolution layers are followed by a max-pooling layer with a (2,2) kernel size. The **crash detection head** is composed of 7 consecutive convolution layers in which the first 6 layers’ kernel sizes are alternating between (3,3), and (1,1), and the applied filters are also alternating between 512 and 256 filters, as shown in Figure 4. However, the last convolution layer is a 1D convolution, kernel size (1,1), with 18 filters reshaped to (2,9) in which 9 regression values correspond to 1 for confidence, 3 for center point (x, y, z), 3 for dimensions (L, W, D), and 2 for two classes ($C_{0}$ or $C_{1}$). A Sigmoid activation function is finally applied to output the normalized continuous values for the crash/no crash bounding box and center with their confidence values. The **action control head** starts with receiving speed input from the CARLA simulator passed through 2 fully connected layers with 128 neurons each. Then, the output from the feature extraction part is flattened and passed through 2 fully connected layers with 256 neurons each. A concatenation layer is applied to the outcome from the extracted features from CARLA speed information and the feature extraction; after that, 2 fully connected layers with 256 neurons are applied to relate the extracted features together. The last 2 fully connected layers with 10 and 3 neurons, respectively, are applied with a Sigmoid activation function layer to output the normalized continuous values for throttling, braking, and steering between 0 and 1. Table 1 shows each layer with the output shape, the number of parameters per layer, and the total number of parameters.

As SDC-Net++ design follows the multitask (two heads) networks; the total loss function for the network will be a combined loss between the two heads (loss from the action control part and loss for the detection part), as shown in Equation (Equation 1).
(1)$$L_{combined}=L_{control}+L_{detection}$$

The action control loss function is a typical mean square error function, as shown in Equation (Equation 2), where *m* represents the normalized samples whose iterator is *i*, $\hat{Y_{i}}$ is the output prediction from the regression model, $Y_{i}$ is the continuous ground truth labels (throttle, steer, brake) corresponding to the inputs, and MSE is the mean square error. However, the detection loss function combines the 4 components shown in Equation (Equation 3): confidence, bounding box center, bounding box dimensions, and classification components.
(2)$$L_{control}=MSE=\frac{1}{m}\sum_{i=1}^{i=m}{\left(\hat{Y_{i}}-Y_{i}\right)}^{2}$$
(3)$$L_{detection}=L_{conf}+L_{center}+L_{dimensions}+L_{classification}$$

The 4 loss components functions are shown in Equations (Equation 4)–(Equation 7), where $\lambda_{conf}$, $\lambda_{coor}$, and $\lambda_{class}$ are the weights for confidence prediction, coordinates, and probabilities of the classifications, respectively, $S^{2}$ is the (S×S) grid cells in which the images are decomposed whose iterator is *i*, *B* bounding boxes are predicted in each grid cell whose iterator is *j*, $I_{ij}^{obj}$ is a binary variable that takes values (0,1) as per the ground truth box in $i^{th}$, $j^{th}$ is the location in which 1 in case of there is a box and 0 is otherwise, $I_{ij}^{Noobj}$ is always the opposite of $I_{ij}^{obj}$, $C_{i}$ and $\hat{C_{i}}$ are the predicted and ground truth of the confidence, $\left(x_{i},y_{i},z_{i}\right)$ and $\left({\hat{x}}_{i},{\hat{y}}_{i},{\hat{z}}_{i}\right)$ are the predicted and ground truth of the box center in the 3D coordinates, $\left(w_{i},l_{i},d_{i}\right)$ and $\left({\hat{w}}_{i},{\hat{l}}_{i},{\hat{d}}_{i}\right)$ are the predicted and ground truth of the box dimensions (width, length, depth), and $p_{i}\left(c\right)$, and ${\hat{p}}_{i}\left(c\right)$ are the predicted and ground truth probabilities of the classes.
(4)$$L_{conf}=\lambda_{conf}\sum_{i=0}^{S^{2}}\sum_{j=0}^{B}I_{ij}^{obj}{\left(C_{i}-\hat{C_{i}}\right)}^{2}+\lambda_{conf}\sum_{i=0}^{S^{2}}\sum_{j=0}^{B}I_{ij}^{Noobj}{\left(C_{i}-\hat{C_{i}}\right)}^{2}$$
(5)$$L_{center}=\lambda_{coor}\sum_{i=0}^{S^{2}}\sum_{j=0}^{B}I_{ij}^{obj}\left[{\left(x_{i}-\hat{x_{i}}\right)}^{2}+{\left(y_{i}-\hat{y_{i}}\right)}^{2}+{\left(z_{i}-\hat{z_{i}}\right)}^{2}\right]$$
(6)$$L_{dimensions}=\lambda_{coor}\sum_{i=0}^{S^{2}}\sum_{j=0}^{B}I_{ij}^{obj}\left[{\left(\sqrt{w_{i}}-\sqrt{\hat{w_{i}}}\right)}^{2}+{\left(\sqrt{l_{i}}-\sqrt{\hat{l_{i}}}\right)}^{2}+{\left(\sqrt{d_{i}}-\sqrt{\hat{d_{i}}}\right)}^{2}\right]$$
(7)$$L_{classifications}=\lambda_{class}\sum_{i=0}^{S^{2}}\sum_{j=0}^{B}I_{ij}^{obj}\sum_{c\in classes}{\left(p_{i}\left(c\right)-\hat{p_{i}}\left(c\right)\right)}^{2}$$

### 3.2. Input Representations and IoT Integration

SDC-Net uses different input representations to formulate a comparative study and identify which of them is the best one. These input representations are front camera view, panorama with either normal or equirectangular stitching view, and bird’s eye view. Front camera view means that the SDC-Net takes the front camera image only from the camera cocoon as an input image to the system; however, panorama views take the four camera images of the cocoon and then either stitch them together or tilt the edges of every image to isolate between them. Bird’s eye view depends on the intrinsic and extrinsic camera calibration parameters to warp the four cocoon images forming the BEV shape. SDC-Net++ keeps the same input representations to compare with the SDC-Net results.

SDC-Net uses a trigger-action digital automation platform like Zapier, IFTTT, or Integromat, so in case of an accident (trigger), the system provides connected vehicles with the accident information via IoT. SDC-Net is also integrated with Valeo’s standardized platform specific for automotives that uses digital gate and let’s do (LDO) platforms. SDC-Net++ also maintains the same IoT solution to share the relevant information about the accident but with the location of the accident (which lane) as additional information. The additional information about the accident location comes out of the modification of changing the classification network to be a detection network so that the accident lane could be identified easily.

## 4. Dataset Setup

As with SDC-Net, due to the lack of datasets that serve our multitask system of controlling vehicle action and detecting crashes, we created our dataset using the CARLA simulator. SDC-Net++ uses exactly the same dataset proposed by SDC-Net to achieve a fair comparative study, so this section tackles the modifications made to the dataset labels.

### 4.1. CARLA Adaptations

SDC-Net is implemented to have a classification network that can identify if the scene contains an accident or not, so the constructed dataset labels were a binary classification: (0, 1) mean no accident and there is an accident, respectively. However, the SDC-Net++ detection network needs the bounding boxes generated from the CARLA simulator and then applies some processing over them to save the crash label. The flowchart in Figure 5 shows the accident detection label, and the steps are also described as follows.

Filter the objects by keeping only the concerned objects such as vehicles, pedestrians, etc.;Loop over all the bounding box centers received from the CARLA simulator;Calculate the distances between bounding box centers;Check if the distances are greater than the threshold tunable distance. If yes, no crash label is applied; however, if no, this means that we have two or more vehicle centers in close proximity to each other;Adapt the bounding boxes information to the plotly [37] python library to check if there are two intersecting boxes;Check if the boxes are intersecting. If yes, unify the two intersected bounding boxes in one box throughout the Min–Max boxes’ boundaries as shown in Figure 6. If no, save the bounding boxes.

### 4.2. Data Analysis

Our dataset is collected using the CARLA simulator. The total number of samples is **125 K** samples in which each sample consists of 4 synchronized images: front, left, right, and rear. Around **62.5%** of the samples are used as training data, **10.7%** are used for validation, and **26.8%** are used for testing as shown in Figure 7. Training data are decomposed into **56.2%** with no crashes and **43.8%** are composed with crash samples, as shown in Figure 8. Validation data are decomposed into **60%** with no crash and **40%** with a crash as shown in Figure 9. Testing data are decomposed into **50%** with and without crashes, as shown in Figure 10.

## 5. Results

As with SDC-Net, the conducted experiments are decomposed into two variants: different input view and single-task network or multitask output. The first variant (input view) is categorized into different views: front camera, panorama, and BEV views; however, the second variant is categorized into running using a single/multiple output head network. As long as SDC-Net++ replaces the classification network with a detection one, the comparative study will tackle only crash avoidance only, and the multitask network will tackle crash avoidance with path planning and AEB, because there is no need to run the path planning and AEB only, because it will be identical to the experiments conducted in SDC-Net. This section covers the experiments conducted in SDC-Net to compare the different input representations as shown in Table 2 in which the rows show the different input representations and columns show single/multitask networks. The main measurement metrics reported are identical to SDC-Net: precision, recall, f1-score, accuracy for the detection network, and mean square error (MSE) in the control action regression.

### 5.1. Crash Detection Only Results

The crash avoidance-only experiment is a single-head detection output network in which the experiment is conducted three times: the first uses the front camera only, the second uses the panorama view, and the last one uses the BEV. Due to the detection problem, the measurement metrics can be intersection over union (IoU); however, to compare with SDC-Net, the measurement metrics are only precision, recall, f1-score, and accuracy. It is obvious that the BEV results are better than the panorama camera and front camera results in precision by 15.197% and 33.203%, respectively; in recall by 15.273% and 44.374%, respectively; in f1-score by 15.246% and 38.867%, respectively; and in accuracy by 5% and 14%, respectively.

### 5.2. Multitask Results

Crash avoidance, path planning, and AEB experiments are multi-head detection–regression output networks (multitask) in which the experiments are conducted three times: the first uses the front camera only, the second uses the panorama view, and the last one uses the BEV. Due to the multitask problem, the measurement metrics are the already previously mentioned ones of precision, recall, f1-score, and accuracy, in addition to the control actions MSE. It is obvious that the BEV results are better than the panorama camera (experiments conducted in SDC-Net) and front camera results in precision by 15.134% and 27.212%, respectively; in recall by 12.046% and 28.706%, respectively; in f1-score by 13.593% and 27.962%, respectively; in accuracy by 5% and 13%, respectively; in throttle MSE by 32.97% and 48.835%, respectively; in steer MSE by 29.173% and 51.624%, respectively; and in brake MSE by 36.578% and 55.924%, respectively.

## 6. Discussion

This technical discussion section is decomposed into two sub-sections: the first sub-section tackles the SDC-Net++ results only; however, the second sub-section tackles the comparison between SDC-Net and SDC-Net++ experimental results.

### 6.1. SDC-Net++ Results Discussion

Regarding **crash avoidance-only experiments**, the front camera-only experiments detect the accidents in front of the ego-vehicle only; however, it is not able to detect accidents beside and behind. This is due to the absence of sensors that can visually see behind and beside the ego vehicle, so it was expected that the measurement metrics were not so good. Panorama and BEV experiments have better results because depending on the camera cocoon setup that covers 360° around the ego vehicle, in case of an accident that happened beside or behind the vehicle, the sensors will be able to visually see it. **Why did the BEV experiments outperform the panorama experiments?** Panorama depends on stitching the raw cocoon images together; however, BEV depends on warping the cocoon images as a pre-processing phase before input to the network. BEV indirectly includes the distance factor because of warping; this facilitates detecting collision.

Regarding **path planning and AEB experiments**, front camera camera-only experiments take control action according to the front image only; however, it fails because it visually cannot see other vehicles besides the ego vehicle, so it takes inaccurate control actions (unsafe) that were expected due to lack of information about the environment. Panorama and BEV experiments have better results due to the different camera cocoon setups that cover 360° around the ego vehicle so that in the case of another vehicle in addition to the ego vehicle, the ego vehicle will not be able to maneuver. **Again, we must ask why did the BEV experiments outperform the panorama experiments?** BEV depends on warping the cocoon images in which the warped front image becomes the upper part of the total image, the warped left image becomes the left part of the total image, the warped right image becomes the right part of the total image, and the rear warped image becomes the lower part of the total image. This setup facilitates extracting the environment features compared to stitching the raw cocoon images together as in the panorama view.

The multitask network outperforms the single task in both detection and regression measurement metrics. Figure 11 shows the bar charts for precision, recall, f1-score, and accuracy; multitask with the BEV is the best experiment metric. Figure 12 shows the bar charts for the MSE of throttle, brake, and steer for the conducted experiments; also, multitask with BEV is the best lower experiment metric. Conceptually, multitask networks prove their efficiency compared with single-task ones because the presence of two or more tasks supports each other to learn in a better way.

### 6.2. SDC-Net and SDC-Net++ Results Discussion

Table 3 provides the combined experimental results for both SDC-Net and SDC-Net++. Regarding **crash avoidance only experiments** with **front camera** input, it is obvious that SDC-Net++ outperforms SDC-Net in precision by 2.702%, recall by 6.203%, f1-score by 4.538%, and accuracy by 2%; with **panorama** view input, SDC-Net++ outperforms SDC-Net in precision by 5.291%, recall by 5.794%, f1-score by 5.549%, and accuracy by 4%; and with **BEV** input, SDC-Net++ also outperforms SDC-Net in precision by 6.135%, recall by 9.694%, f1-score by 7.951%, and accuracy by 5%. Regarding **multitask** with **front camera** input, it is obvious that SDC-Net++ outperforms SDC-Net in precision by 9.090%, recall by 17.916%, f1-score by 13.55%, and accuracy by 2%; with **panorama** view input, SDC-Net++ outperforms SDC-Net in precision by 2.016%, recall by 2.678%, f1-score by 2.344%, and accuracy by 3%; and with **BEV** input, SDC-Net++ also outperforms SDC-Net in precision by 2.201%, recall by 2.8%, f1-score by 2.505%, and accuracy by 2%. Regarding **multitask** with **front camera** input, it is also obvious that SDC-Net++ outperforms SDC-Net in throttle MSE by 6.673%, steer MSE by 1.314%, and brake MSE by 3.691%; with **panorama** view input, SDC-Net++ outperforms SDC-Net in throttle MSE by 11.275%, steer MSE by 9.335%, and brake MSE by 9.613%; and with **BEV** input, SDC-Net++ also outperforms SDC-Net in throttle MSE by 18.678%, steer MSE by 16%, and brake MSE by 21.353%.

SDC-Net++ outperforms SDC-Net in both detection and regression measurement metrics. Figure 13 shows the bar charts for precision, recall, f1-score, and accuracy for both SDC-Net and SDC-Net in the case of crash avoidance only. Figure 14 shows the bar charts for the MSE of throttle, brake, and steer for both SDC-Net and SDC-Net++. Thus, Figure 15 shows the bar charts for precision, recall, f1-score, and accuracy for both SDC-Net and SDC-Net in the case of multitask networks. Conceptually, multitask networks prove their efficiency compared with single-task ones because the presence of two or more tasks supports each other to learn in a better way. Finally, the SDC-Net++ detection network enhances the measurement metrics compared with the classification one in SDC-Net for the following reasons:Directing the labels to detect where exactly the accident (bounding box) is in the scene rather than the classification that has only a binary label for the scene if there is an accident or not.Enlarging the size of the network and thus enlarging the trainable parameters, which means that the SDC-Net design was not able to extract the most important features (accident features) in the input scene.

## 7. Conclusions

Currently, self-driving cars harness the power of the artificial intelligence (AI). Self-driving cars benefit from sensor fusion to perceive and understand the surrounding environment. IoT plays a very vital role in functioning self-driving cars, as it connects vehicles wirelessly sharing environment on-board sensors’ information. Autonomous vehicles utilize this connectivity to update the driving algorithms as a sort of incremental enhancement. Our proposed system, SDC-Net++, which is an end-to-end crash detection and action control self-driving car system, is proposed to enhance our prior study of the SDC-Net system, which tackles exactly the same problem. SDC-Net++ replaced the classification network with a detection one, keeping the same benchmark dataset to achieve a fair comparative study. SDC-Net++ with BEV input outperforms the nearest representation in precision, recall, f1-score, and accuracy by more than 15.134%, 12.046%, 13.593%, and 5%, respectively. It also outperforms SDC-Net in precision, recall, f1-score, accuracy, and average MSE by more than 2.201%, 2.8%, 2.505%, 2%, and 18.677%, respectively.

## Figures and Tables

**Figure 1 sensors-24-03805-f001:**
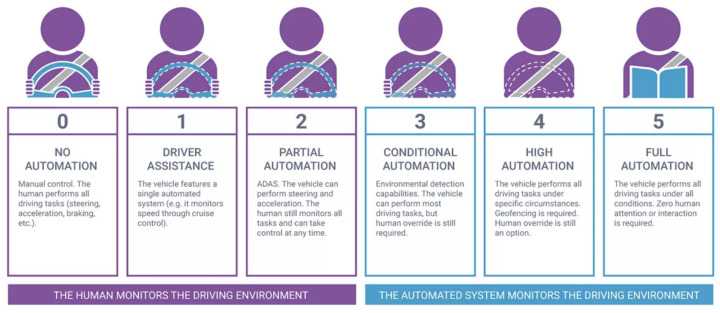
Autonomous driving levels.

**Figure 2 sensors-24-03805-f002:**
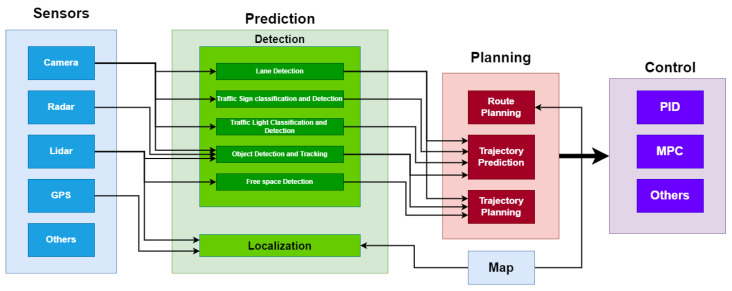
Self-driving pipeline.

**Figure 3 sensors-24-03805-f003:**
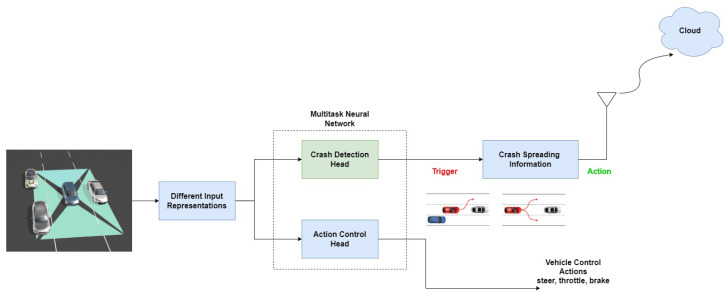
System architecture.

**Figure 4 sensors-24-03805-f004:**
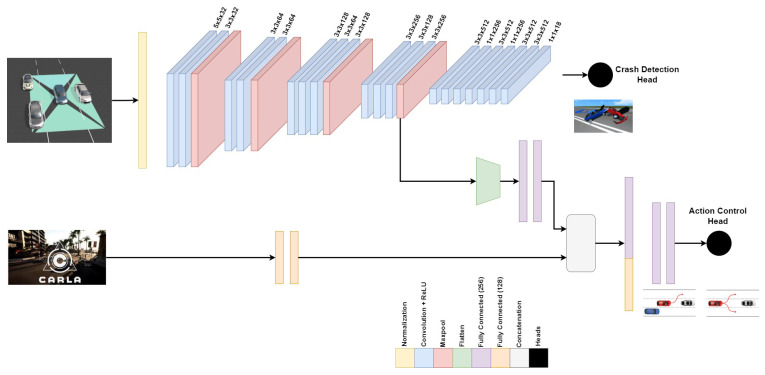
SDC-Net++: multitask deep neural network.

**Figure 5 sensors-24-03805-f005:**
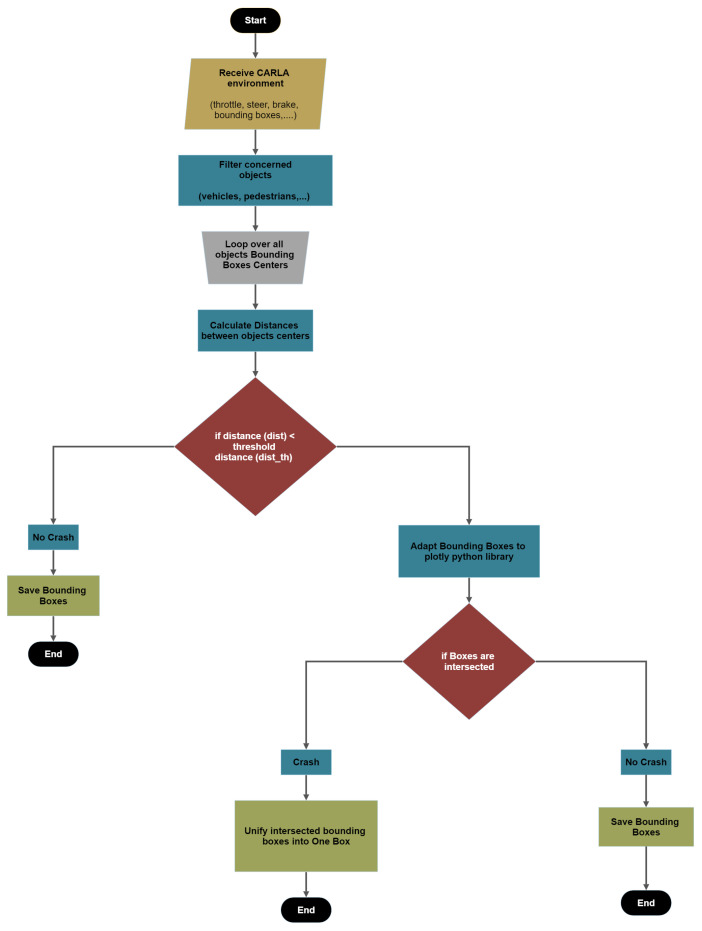
Crash labels processing flowchart.

**Figure 6 sensors-24-03805-f006:**
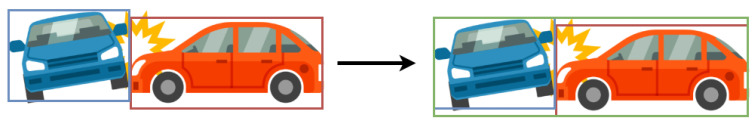
Unified boxes.

**Figure 7 sensors-24-03805-f007:**
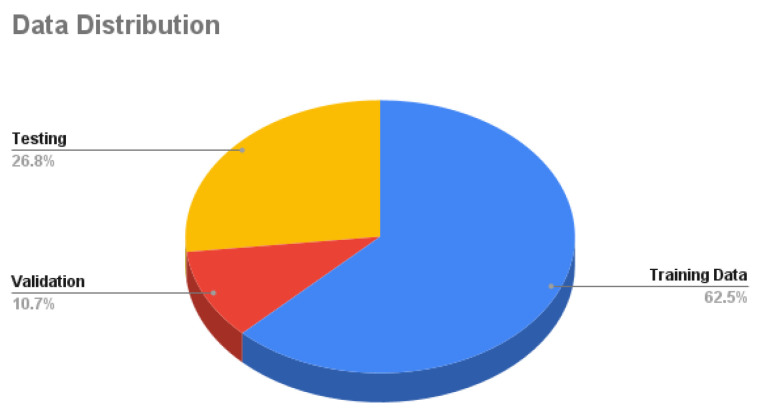
Dataset distribution.

**Figure 8 sensors-24-03805-f008:**
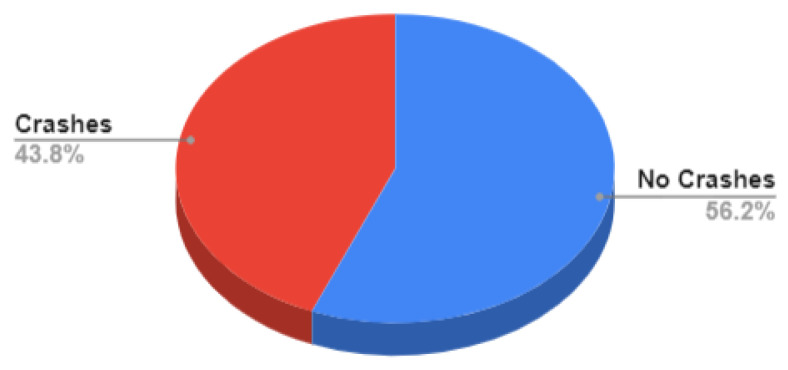
Training Data.

**Figure 9 sensors-24-03805-f009:**
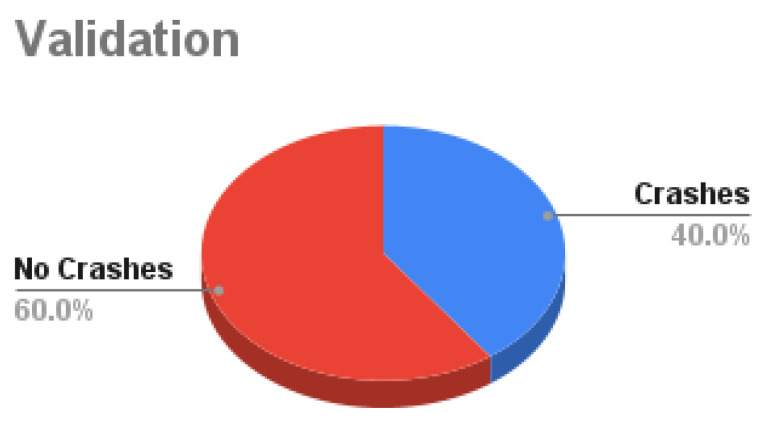
Validation Data.

**Figure 10 sensors-24-03805-f010:**
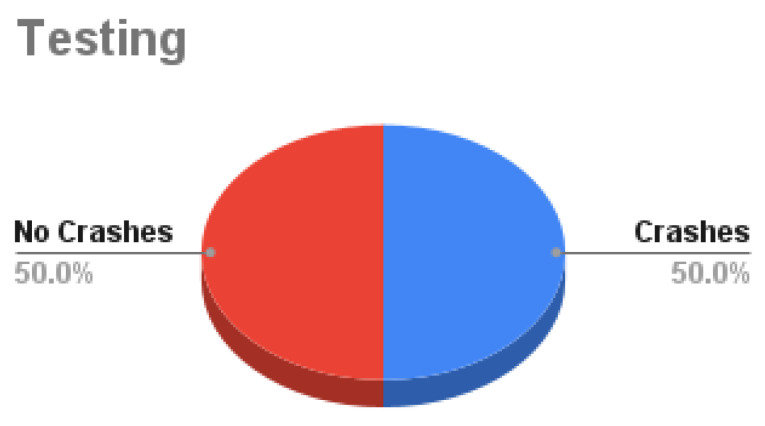
Testing Data.

**Figure 11 sensors-24-03805-f011:**
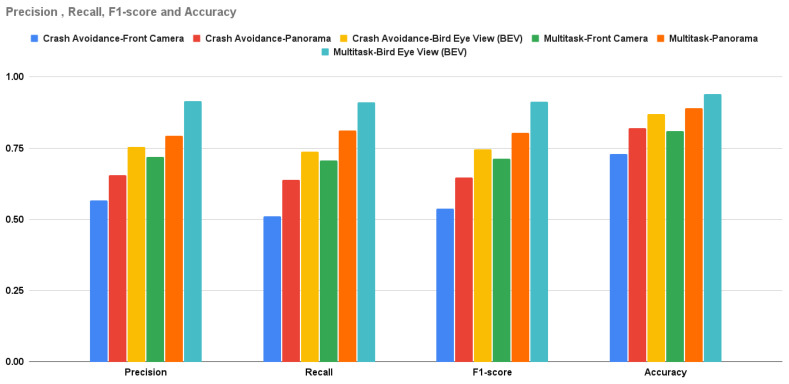
Precision, recall, f1-score, and accuracy for crash avoidance only vs. multitask.

**Figure 12 sensors-24-03805-f012:**
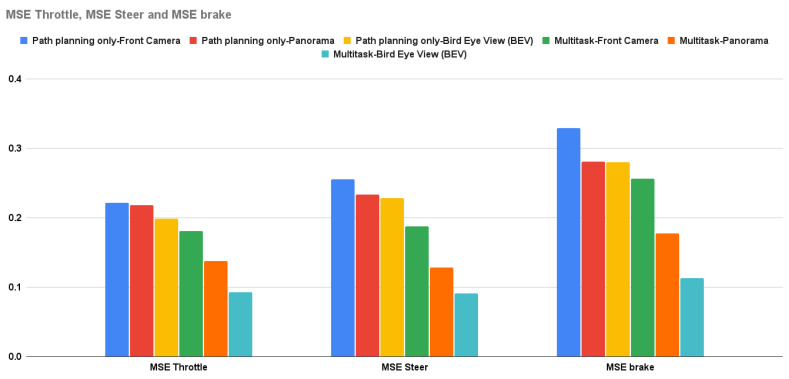
MSE throttle, steer, and brake for path planning, AEB only vs. multitask.

**Figure 13 sensors-24-03805-f013:**
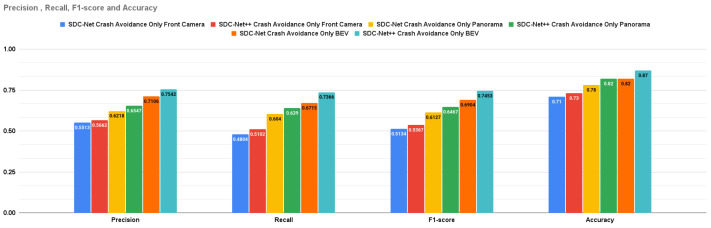
Detection metrics for SDC-Net vs. SDC-Net++ with crash avoidance only.

**Figure 14 sensors-24-03805-f014:**
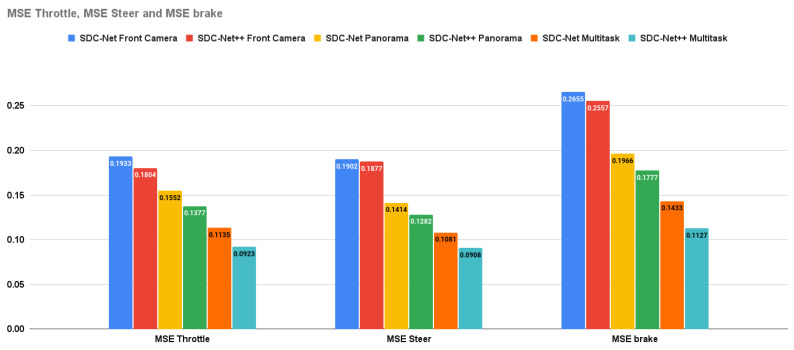
Regression metrics for SDC-Net vs. SDC-Net++.

**Figure 15 sensors-24-03805-f015:**
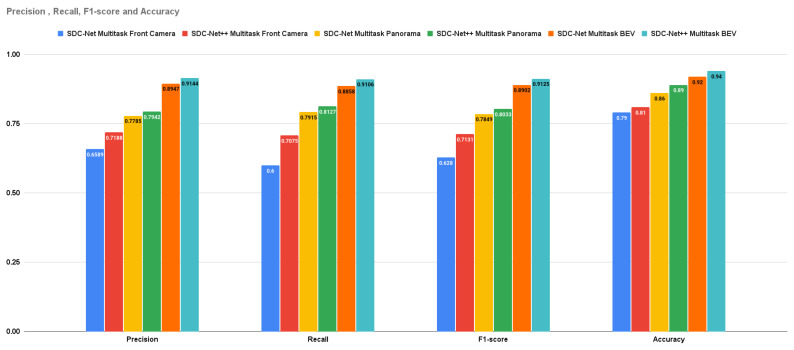
Detection metrics for SDC-Net vs. SDC-Net++ with multitask.

**Table 1 sensors-24-03805-t001:** SDC-Net++ network parameters.

Layer	Output Shape	Number of Parameters
Input Layer	(None, 120, 300, 3)	-
Conv1	(None, 120, 300, 32)	2432
Conv2	(None, 120, 300, 32)	9248
MaxPool1	(None, 60, 150, 32)	-
Conv3	(None, 60, 150, 64)	18,496
Conv4	(None, 60, 150, 64)	36,928
MaxPool2	(None, 30, 75, 64)	-
Conv5	(None, 30, 75, 128)	73,856
Conv6	(None, 30, 75, 64)	73,792
Conv7	(None, 30, 75, 128)	73,856
MaxPool3	(None, 15, 38, 128)	-
Conv8	(None, 15, 38, 256)	295,168
Conv9	(None, 15, 38, 128)	295,040
Conv10	(None, 15, 38, 256)	295,168
MaxPool4	(None, 8, 19, 256)	-
Crash Head (Conv11)	(None, 8, 19, 512)	1,180,160
Crash Head (Conv12)	(None, 8, 19, 256)	131,328
Crash Head (Conv13)	(None, 8, 19, 512)	1,180,160
Crash Head (Conv14)	(None, 8, 19, 256)	131,328
Crash Head (Conv15)	(None, 8, 19, 512)	1,180,160
Crash Head (Conv16)	(None, 8, 19, 512)	2,359,808
Crash Head (Conv17)	(None, 8, 19, 18)	9234
Crash Head(Reshape)	(None, 8, 19, 2, 9)	-
Flatten	(None, 38,912)	-
FC1-256	(None, 256)	9,961,728
FC2-256	(None, 256)	65,792
Input Speed	(None, 1)	-
FC1-128	(None, 128)	256
FC2-128	(None, 128)	16,512
Concat (FC2-256,FC2-128)	(None, 384)	-
FC3-256	(None, 256)	98,560
FC4-256	(None, 256)	65,792
Control Head (FC1-10)	(None, 10)	2570
Control Head (FC2-3)	(None, 3)	33
Sigmoid	(None, 3)	-
		17,557,405

**Table 2 sensors-24-03805-t002:** Measurement Metrics Comparison.

		Experiments													
		**Crash Avoidance** **Only**				**Path Planning** **and AEB** **Only**			**Crash Avoidance,** **Path Planning and** **AEB**						
**Input** **Representations**		Precision	Recall	F1-Score	Accuracy	MSE throttle	MSE Steer	MSE brake	Precision	Recall	F1-Score	Accuracy	MSE throttle	MSE Steer	MSE brake
	Front Camera	0.5662	0.5102	0.5367	0.73	0.2214	0.2552	0.3285	0.7188	0.7075	0.7131	0.81	0.1804	0.1877	0.2557
	Panorama	0.6547	0.639	0.6467	0.82	0.2176	0.2333	0.2805	0.7942	0.8127	0.8033	0.89	0.1377	0.1282	0.1777
	Bird’s Eye View (BEV)	0.7542	0.7366	0.7453	0.87	0.1988	0.2279	0.2794	**0.9144**	**0.9106**	**0.9125**	**0.94**	**0.0923**	**0.0908**	**0.1127**

**Table 3 sensors-24-03805-t003:** SDC-Net vs. SDC-Net++.

		Experiments													
		**Crash Avoidance** **Only**				**Path Planning and** **AEB Only**			**Multitask**						
**SDC-Net**		**Precision**	**Recall**	**F1-Score**	**Accuracy**	**MSE Throttle**	**MSE Steer**	**MSE Brake**	**Precision**	**Recall**	**F1-Score**	**Accuracy**	**MSE Throttle**	**MSE Steer**	**MSE Brake**
	**Front Camera**	0.5513	0.4804	0.5134	0.71	0.2214	0.2552	0.3285	0.6589	0.6	0.628	0.79	0.1933	0.1902	0.2655
	**Panorama**	0.6218	0.604	0.6127	0.78	0.2176	0.2333	0.2805	0.7785	0.7915	0.7849	0.86	0.1552	0.1414	0.1966
	**BEV**	0.7106	0.6715	0.6904	0.82	0.1988	0.2279	0.2794	**0.8947**	**0.8858**	**0.8902**	**0.92**	**0.1135**	**0.1081**	**0.1433**
**SDC-Net++**	**Front Camera**	0.5662	0.5102	0.5367	0.73	-	-	-	0.7188	0.7075	0.7131	0.81	0.1804	0.1877	0.2557
	**Panorama**	0.6547	0.639	0.6467	0.82	-	-	-	0.7942	0.8127	0.8033	0.89	0.1377	0.1282	0.1777
	**BEV**	0.7542	0.7366	0.7453	0.87	-	-	-	**0.9144**	**0.9106**	**0.9125**	**0.94**	**0.0923**	**0.0908**	**0.1127**

## Data Availability

Data are contained within the article.

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
