# Peer review of "SDC-Net++: End-to-End Crash Detection and Action Control for Self-Driving Car Deep-IoT-Based System"

_sensors, 2024, doi:10.3390/s24123805_

Round 1

Reviewer 1 Report

Comments and Suggestions for Authors

The present work shows promising advancements achieved by the SDC-Net++ system over the original SDC-Net. Remarkable improvements in precision, F1-score, recall and accuracy have been obtained. The presented outcomes highlight the higher performance of SDC-Net++ in crash detection and IoT-based communication. The research presents a significant contribution to the field of self-driving cars. I think that the paper can be accepted in its present form. I would like to ask the authors just few things:

1. Has the structure of Multitask deep neural network, inspired by [35], as presented in 3.1. Multitask Deep Neural Network, been optimized for the present work?
2. Specifically, have preliminary hyperparameter grid search optimizations been conducted to determine the best structure?
3. If yes, could you please explain the reasons behind choosing such a structure?
4. Moreover, was the evaluation of the best structure based on metrics during the validation process or on the output performance?  

Comments on the Quality of English Language

Minor editing of English language required

Author Response

Kindly find attached my reply for the review comments. 

Reviewer 2 Report

Comments and Suggestions for Authors

The paper targets autonomous driving applications and proposes SDC-Net++, which is an end-to-end crash detection and action control self-driving car system. SDC-Net++ replaces the classification network with a detection one in prior SDC-Net system and is able to use BEV input to enhance its performance. The evaluations are implemented on the same benchmark dataset with SDC-Net system and the results demonstrate that SDC-Net++ with BEV input outperforms the nearest representation in terms of precision, recall, F1-score and accuracy. Also,  SDC-Net++ outperforms SDC-Net.

Overall, the paper targets a practical problem and proposes an improved solution. Also, the evaluations are comprehensive. However, there is a few comments below and it is advised that the authors can address it.

Compared to SDC-Net, SDC-Net++ replaces classification network with detection network. SDC-Net++  can only focus on crash-avoidance and multi-tasks application compared to SDC-Net, though SDC-Net++ outperforms SDC-Net if it uses BEV input. This way, does it mean SDC-Net++ causes function regression compared SDC-Net? Besides, is there any chance that the authors can discuss the practical deployment of SDC-Net++ in terms of resources utilization?

Comments on the Quality of English Language

Looks good to me. 

Author Response

Kindly find attached my reply to the review comments.
